# Segmentation-Based vs. Regression-Based Biomarker Estimation: A Case Study of Fetus Head Circumference Assessment from Ultrasound Images

**DOI:** 10.3390/jimaging8020023

**Published:** 2022-01-25

**Authors:** Jing Zhang, Caroline Petitjean, Samia Ainouz

**Affiliations:** Normandie University, INSA Rouen, UNIROUEN, UNIHAVRE, LITIS, 76000 Rouen, France; caroline.petitjean@univ-rouen.fr (C.P.); samia.ainouz@insa-rouen.fr (S.A.)

**Keywords:** head circumference, ultrasound imaging, segmentation CNN, regression CNN, biomarker estimation

## Abstract

The fetus head circumference (HC) is a key biometric to monitor fetus growth during pregnancy, which is estimated from ultrasound (US) images. The standard approach to automatically measure the HC is to use a segmentation network to segment the skull, and then estimate the head contour length from the segmentation map via ellipse fitting, usually after post-processing. In this application, segmentation is just an intermediate step to the estimation of a parameter of interest. Another possibility is to estimate directly the HC with a regression network. Even if this type of segmentation-free approaches have been boosted with deep learning, it is not yet clear how well direct approach can compare to segmentation approaches, which are expected to be still more accurate. This observation motivates the present study, where we propose a fair, quantitative comparison of segmentation-based and segmentation-free (i.e., regression) approaches to estimate how far regression-based approaches stand from segmentation approaches. We experiment various convolutional neural networks (CNN) architectures and backbones for both segmentation and regression models and provide estimation results on the HC18 dataset, as well agreement analysis, to support our findings. We also investigate memory usage and computational efficiency to compare both types of approaches. The experimental results demonstrate that even if segmentation-based approaches deliver the most accurate results, regression CNN approaches are actually learning to find prominent features, leading to promising yet improvable HC estimation results.

## 1. Introduction

The automated measurement of fetus head circumference (HC) is performed throughout pregnancy as a key biometric to monitor fetus growth and estimate gestational age. In a clinical routine, this measurement is performed on ultrasound (US) images, via manually tracing the skull contour, along to fitting it to an ellipse, which is done by sonographers. Figure 1 is one sample of an ultrasound (US) image of fetus head from the HC18 public dataset [1] used in this paper. Identifying the head contour is challenging due to low signal-to-noise ratio in US images, and also because the contours have fuzzy (and sometimes missing) borders (Figure 1). Manual contouring is an operator-dependant operation, subject to intra- and inter-variability, which yields inaccurate measurements, as measured in [2]: The 95% limits of agreement have been measured to ±7 mm for the intra-operator variability and ±12 mm for the inter-operator variability.

Usually, automating the measurement of fetus head circumference in US images is achieved through automatic segmentation methodology. Segmentation methods typically involves image-processing or machine learning-based approaches, some post-processing of the result, so as to fit it into an ellipse. This process involves multiple steps, is adhoc, and can be prone to error. Let us emphasize on the fact that here, the segmentation is just an intermediate step to compute a characteristic from the image, i.e., the length of the head contour. More generally, segmentation is often a prerequisite step toward the computation of biomarkers: For example, the cardiac ventricles are segmented in magnetic resonance images in order to estimate the cardiac contractile function via some indices (e.g., ejection fraction) [3]. Another example is anthropometry, where measuring the skeletal muscle body mass and fat body mass, which are a significant prognostic factors in cancer, are estimated from the segmentation of muscle and fat in computed tomography (CT) images [4]. Instead of resorting to segmentation, which is a costly and error-prone process, one can attempt to estimate the (single or multiple) characteristics or biomarkers, directly. Works on this topic have gotten a second wind with the breakthrough of deep learning, which allows one to take advantage of the power of feature representation and to perform an end-to-end regression [5,6,7,8,9,10].

However direct, “segmentation-free” approaches rely on much less information to estimate the biomarker, and it is not clear yet if segmentation-free approaches can reach the level of accuracy of segmentation-based approaches. To our knowledge, there is no study that has rigorously compared segmentation-based methods and segmentation-free methods for a given application of biomarker estimation, and quantifies the gap between them. This observation motivates the present study, where we propose a fair, quantitative comparison of segmentation-based and segmentation-free (i.e., regression) approaches to estimate how far regression-based approaches stand from segmentation approaches, for the estimation of the head circumference in US images. More precisely, we investigate several settings, i.e., state-of-the-art segmentation models and various backbones for the regression CNN architectures, to obtain the best of both worlds, and to also investigate time and memory consumption in addition to estimation accuracy.

The rest of the paper is organized as follows. Section 2 describes the related segmentation-based works about HC estimation from US images on one side and segmentation-free approaches for biomarker estimation on the other side. Section 3 introduces the methodological framework for segmentation-based and segmentation-free methods. In Section 4 we describe the dataset, the experiment protocol. We present and discuss the experimental results comparing both the segmentation-based and segmentation-free approaches in Section 5. At last, the conclusion is drawn in Section 6.

## 2. Related Works

### 2.1. Fetus Head Circumference Estimation

Several approaches have been proposed in the literature to measure the fetus head circumference in US images, based on image segmentation [11,12,13]. Some follow a two-step approach, namely fetus head localization and segmentation refinement [11]. For example, in [14], the first step consists of locating the fetus head with Haar-like features used to train a random forest classifier; and the second step consists of the measurement of the HC, via ellipse fitting and Hough transform. Other approaches build upon deep segmentation models also in a two-step process, contour prediction, and ellipse fitting [15]. In [16], the standard segmentation model U-Net [17] is trained using manually labeled images, and segmentation results are fitted to ellipses. In [18], authors build upon the same idea, using the multi-task learning paradigm to jointly segment the US image and estimate the ellipse parameters. In [19], the authors use first a region-proposal CNN for head localization, and a regression CNN trained on distance fields to segment the HC. Ref. [20] advances the work [19] since they propose a Mask-R^2^CNN neural network to perform HC distance-field regression for head delineation in an end-to-end way, which does not need prior HC localization or postprocessing for outlier removal. All these methods rely on a segmentation of the fetus head as a prerequisite to estimating the HC.

### 2.2. Segmentation-Free Approaches for Biomarker Estimation

Works aimed at directly extracting biomarkers from medical images have gained traction these last years, especially thanks to advances in deep learning. The goal is to avoid intermediate steps, such as segmentation and other adhoc post-processing steps, that maybe computationally expensive (for both model training and images annotation) and prone to errors [5]. Direct parameter estimation with deep learning can be found in various medical applications; for example, in [5], the authors propose a learning-based approach to perform a direct volume estimation of the cardiac left and right ventricles from magnetic resonance (MR) images, without segmentation. The approach consists in computing shape descriptors using a bag-of-word model, and to perform Bayesian estimation with regression forests. Ref. [6] utilizes regression forest to directly estimate the kidney volume on computed tomography images. Ref. [7] quantify spine indices from MRI via regression CNN with feature amplifiers. Ref. [8] propose multi-task learning for the measurement of cardiac volumes from MRI. For vascular disease diagnosis, Ref. [9] quantify 6 indices of the coronary artery stenosis from X-ray images by using multi-output regression CNN model with an attention mechanism. Preliminary results on the estimation of the head circumference in US images with regression CNN are presented in [10]. By taking advantage of the representation power of CNN, one can now skip the feature design step and learn the features, while at the same time performing the estimation of the value of interest, i.e., regression. Regression CNN are also at the heart of other fields in computer vision, such as head-pose estimation [21], facial landmark detection [22], and human-body pose estimation [23].

## 3. Methodological Framework

### 3.1. Head Circumference Estimation Based on Segmentation

#### 3.1.1. CNN Segmentation Model

We investigate several segmentation architectures that are the state-of-the-art network in medical image segmentation, to segment the contour of a fetus head: The well-known U-Net model [17], U-Net++ [24], DoubleU-Net [25], FPN [26], LinkNet [27], and PSPNet [28]. We trained these architectures from scratch but also investigate transfer learning as a way to mitigate the limited number of images in the HC18 dataset. Even though the natural images from ImageNet and US images have obvious dissimilarities, some generic representations can be learned from a large-scale dataset that might be beneficial to other types of images, and they have proven so in the context of MR images [29]. Thus we have used various backbone models, namely VGG16 [30], ResNet50 [31], and EfficientNet [32], pre-trained on the ImageNet dataset, for all architectures mentioned above. For the loss function, we use the Dice loss, highlighted by [33] to be one of the best loss functions for medical image segmentation.

#### 3.1.2. Post-Processing of Segmentation Results

It can happen that the segmentation results have some noise or incomplete part such as holes, which can cause inaccurate ellipse fitting. Thus some post-processing is applied on the segmentation results: Contours are detected from the segmentation map by the Canny filter, then the largest connected component is kept.

#### 3.1.3. HC Computation Based on Segmentation Results

After post-processing the segmented results, the next step is to perform ellipse fitting in order to get the parameters (long axis, short axis, center points, angle) of the ellipse to compute its length. The length of an ellipse denoted HC is approximated by the Ramanujan approximation method [34] in which $h=\frac{{\left(a-b\right)}^{2}}{{\left(a+b\right)}^{2}}$, *a* and *b* being the long and short axis of the ellipse:(1)$$\mathrm{HC}=\pi \left(a+b\right)\left(1+\frac{3h}{10+\sqrt{4-3h}}\right).$$

### 3.2. Head Circumference Estimation Using Regression CNN

#### 3.2.1. Regression CNN Model

As shown in Figure 2, the regression CNN are composed of a CNN backbone and regression layer (linear activation function), which can learn the features of the input fetus head to estimate the HC value directly. The backbone CNN that we experimented with are state-of-the-art architectures: VGG16 [30], ResNet50 [31], EfficientNetb2 [32], DenseNet121 [35], Xception [36], MobileNet [37], and InceptionV3 [38]. In order to improve model convergence, and for the reasons stated above in the previous section, we use them pretrained on ImageNet [39], and fine-tune them for the task at hand.

#### 3.2.2. Loss Functions

The loss functions commonly used in regression CNN include the mean absolute error (MAE) loss, mean square error (MSE) loss, and the Huber loss (HL), defined as:(2)$$\mathrm{MAE}\;\mathrm{loss}=\frac{1}{N}\sum_{i=1}^{N}\left|p_{i}-g_{i}\right|$$
(3)$$\mathrm{MSE}\;\mathrm{loss}=\frac{1}{N}\sum_{i=1}^{N}{\left(p_{i}-g_{i}\right)}^{2}$$
(4)$$\mathrm{HL}=\left\{\begin{array}{c} \frac{1}{N}\sum_{i=1}^{N}\frac{1}{2}{\left(p_{i}-g_{i}\right)}^{2},\;\;\mathrm{for}\;\left|p_{i}-g_{i}\right|<\delta \\ \frac{1}{N}\sum_{i=1}^{N}\delta *(|p_{i}-g_{i}\left|-\frac{\delta }{2}\right),\;\;\mathrm{otherwise} \end{array}\right.$$
where $p_{i}$ is the probability of predicted pixels, $g_{i}$ is the real value of head circumference in pixels, and *N* is the number of pixels in an image. δ is a hyper parameter and empirically set to 1. We will investigate all three of them, as there is no heuristic to choose one loss over the other, as highlighted in [40].

### 3.3. Model Configuration

For regression models, both the weights of the CNN feature extractor part and regression layer are trainable. As the number of training data of HC18 is limited, to avoid over-fitting, we set the dropout rate as 0.7; in other words, 30% of parameters in regression CNN models are kept. The number of trainable parameters of each model is listed in Table 1.

## 4. Experimental Settings

### 4.1. Dataset and Pre-Processing

The HC18 dataset [1] contains 999 US images acquired during the various trimesters of the pregnancy, along with the corresponding ground truth of the skull contour map and HC values. The reference contour of a fetus head is annotated as an ellipse shape by a professional sonographer and the HC value as well as the pixel size of each image is given in a text file. The gestational age range of this dataset is 10–40 weeks [1].

Image preprocessing includes a resizing from 800 × 540 pixels to 224 × 224, and normalization by subtracting the mean and dividing by standard deviation. The HC values are normalized by dividing by the maximum value of HC, in order to improve convergence. We split the dataset into a training set (600 images), validation set (199 images), and test set (200 images) in a random order. We augment the data of the training set by performing horizontal flipping, and rotation with 10 degrees, the amount of training data is 1800 images.

### 4.2. Experiment Configuration

Both approaches, segmentation or regression, are evaluated with the same protocol, namely with 5-fold cross validation, the folds being identical for all the methods. We set the optimizer as Adam with a learning rate of 10${}^{-4}$. The batch size is 16. The training takes 100 epochs. The implementation is based on Keras, using the public Python library Segmentation Models [41]. Models are trained on the Tesla P100 GPU server with 16 GB of memory.

### 4.3. Evaluation Metrics

Evaluation metrics for the segmentation results are the Dice index (DI), the Hausdorff distance (HD), which is the maximum point-to-point distance (Dist) between two contours, and the average symmetric surface distance (ASSD), between the segmented results (Seg) and the ground truth (GT) map. We use the mean absolute error (MAE) and the percentage MAE (PMAE) to compare the predicted and ground truth HC values:(5)$$Dice=\frac{2Seg\cdot GT}{Seg+GT}$$
(6)HD=Max(Max(Dist(Seg,GT)),Max(Dist(GT,Seg)))
(7)ASSD=Mean(Mean(Dist(Seg,GT)),Mean(Dist(GT,Seg)))
(8)$$PMAE=\frac{MAE}{GT}\cdot 100\%.$$

## 5. Results and Discussion

### 5.1. HC Estimation Based on Segmentation

We train and test 6 different segmentation architectures (U-Net, U-Net++ DoubleU-Net, FPN, LinkNet, and PSPNet) with three CNN backbones (VGG16, ResNet50, and EfficientNet). We found that the segmentation models pretrained on ResNet50 outperformed the other two CNN backbones. So we only report the detailed quantitative evaluation for the ResNet50 backbone, to which we added the original U-Net architecture [17], as shown in Table 2, that contains both the segmentation accuracy and the HC estimation MAE.

From Table 2, one can gather that segmentation-wise, all segmentation models obtained similar scores, as shown by values in columns DI, HD, and ASSD in the Table. However, when it comes to the estimation error of the HC, the U-Net-B2 and LinkNet-B2 are the best architectures, as assessed by a two-sided, paired Student’s *t*-test between the pair of method scores, which resulted in a *p*-value inferior to 0.05 for these two networks. Both networks achieve an MAE value (after post-processing) of 1.08 mm and 1.15 mm, respectively. Post-processing allows one to obtain a small enhancement in the MAE value.

We also analyzed some segmentation results (Figure 3) on some vague US fetus head images; the influence of noise and artifacts of images in segmentation-based methods is less than that in the segmentation-free methods (presented in Figure 4).

### 5.2. HC Estimation Based on Regression CNN

We train and test regression CNN architectures with seven different pretrained CNN backbones, experimenting with three regression loss functions (MAE loss, MSE loss, and Huber loss) on the HC18 dataset. The evaluations of direct HC estimation are given in Table 3. One can find that the Regression EfficientNet (Reg-B3-L1) in conjunction with the MAE loss, performs better than the other CNN models: The resulting MAE for this regression network is 1.83 mm.

### 5.3. Interpretability of Regression CNN Result

#### 5.3.1. Saliency Maps of Regression CNN Results on HC

Contrary to segmentation models, regression models come at a cost of low interpretability, i.e., the model is not providing explicit explanations along with the HC prediction. In order to shed light on what is indeed learned by the regression CNN, we use a post-hoc explanation method to analyze the regression model. In our previous work [42], we showed, with a quantitative perturbation study, that the Layer-wise relevance propagation (LRP) method [43] was appropriate to explain CNN regression models for this application. The idea of LRP is to compute a relevance score for each input pixel layer by layer in a backward direction. It first forward-passes the image so as to collect activation maps and backpropagates the error, taking into account the network weights and activations, yielding saliency maps [44], in which the areas that most contributed to a decision are highlighted. Note that in [45], authors also used the LRP method to explain the results of a regression CNN that aims towards counting leaf on plant photographs. One can discover from Figure 5 that the regression CNN can indeed find the key features from the head contour on the input US images and relies on, to some extent, many contour pixels to make the HC estimation.

#### 5.3.2. Saliency Maps on Outlier Analysis

We also display some saliency maps where regression models fail to make an accurate estimation (see Figure 4). We observe that the features extracted by regression CNN models are fooled by the hypersignal (i.e., high intensity pixels) above the head, which leads to increased predicted HC values. This illustrates the case where the background is heterogeneous and makes it difficult for the network to distinguish the head contour and thus to accurately estimate the head circumference.

### 5.4. Comparison of Segmentation CNN vs. Regression CNN

To compare the performance of the segmentation-free vs. the segmentation approaches, we gathered the two best results from Table 2 and Table 3 into Table 4. From this table, one can see that the best segmentation approach (U-Net-B2: U-Net with pretrained ResNet50 with post-processed segmentation results) is better than the best regression approach (Reg-B3-L1) by 40.7%. We can also notice from Figure 6 that both segmentation and regression methods are correctly fitting the data, the fitting of the segmentation-based method being even smoother.

We also analyze the agreement between the estimated HC values by both types of methods against the real HC values via linear regression. From Figure 7, one can first observe a remarkable linear correlation between the prediction and reference values, for all four models, whether it is segmentation or regression models. There is a tiny fluctuation in regression CNN models in the right top that illustrates that the regression models have a tendency to underestimate the large HC values (this trend will also appear in the Bland–Altman analysis).

The Bland–Altman plot is another way to analyze the agreement between two measurements, by plotting the difference between the measurements vs. their mean, which makes it easy to spot a bias between the measurements. From the Bland–Altman plot in Figure 8, obtained on a fold of 200 test images, we observe that regression approaches struggle with larger fetus head images, which is interesting since segmentation approaches usually fail on small structures. One can also see that for the segmentation models, 8 out of 200 points are outside the 95% agreement limit; for regression models, there are 12 outliers out of 200, mostly distributed in larger HC values. Unsurprisingly, room for improvement is left for regression-based approaches. One can also identify the 95% agreement limits: For the best segmentation model, they are [−3.12 mm, 0.7 mm], and for the best regression model, they are [−3.25 mm, 2.92 mm]. We can compare these limits to the 95% agreement limits on inter-operator variability, which is ±12 mm ([2], Table 1, p. 272): The fact that they are greatly smaller highlights the high relevance of both segmentation-based and segmentation-free approaches as an alternative to automatically estimate the HC from US images. However, the comparison to manual variability should be handled with care as these results have not been obtained on the same dataset.

### 5.5. Memory Usage and Computational Efficiency

The theoretical memory usage of a CNN during training requires to store the network parameters and the activation outputs of every layer, used to compute the gradients, for each batch. As show in Table 5, as one could expect that regression CNN models require less memory storage in general, than the segmentation-based approaches, see column Mem-M. However in practice, the gap between regresssion and segmentation models is not so large, as shown by the actual memory cost in the prediction stage, defined as the maximum used memory when the inference is stable (computed using Python library *Memory Profiler*). In particular, the best regression method (Reg-B3-L1) is requires even more memory than segmentation methods.

As Table 5 shows, the training time per epoch over 1800 training US images for the segmentation method U-Net-B2 (U-Net with ResNet50), takes 29 s on a Tesla P100 GPU. For the best regression model Reg-B3-L1 (EfficientNet), it takes 20 s. In the prediction stage with a Intel Core i7 CPU, 32 GB RAM, the Regression Reg-B3-L1 only takes 36.95 s over 200 test images; in other words, predicting one image requires 0.18 s to be compared to 0.35 s of the U-Net-B2. Segmentation-based methods require longer time at training but also at inference time, than segmentation-free methods. As a conclusion, while the advantage of using regression-based approach is clear computationwise, there is no clear evidence that regression models are less memory greedy, in the experimental conditions we set up. It is worthy to note that with the continuous progress of hardware and computing power, such a time error between segmentation-based and segmentation-free methods may be ignored in clinical practice.

### 5.6. Comparison of HC Estimation with State-of-the-Art

At last, the proposed segmentation-based methods and segmentation-free methods are compared with state-of-the-art (SotA) segmentation methods, which is the standard way of estimating the HC as recalled in Section 2.1. Although a fully accurate comparison is not possible since the experimental protocols are different in each paper (e.g., in [16], the model is trained on the HC18 dataset combined with other fetus head US images), we provide the results as a mean to estimate the order of magnitude of the estimation error. Results given in Table 6 show that the estimation error of the proposed segmentation-based and regression-based approaches based on transfer learning has the same order of magnitude than approaches made of multiple adhoc steps, and dedicated to this task.

## 6. Conclusions and Future Work

In this paper, we addressed the problem of HC estimation from US images via both a conventional segmentation approach with post-processing and ellipse fitting, and a regression-based approach that can directly predict HC without segmentation intervention. Our idea was to quantify how far regression-based approaches stand from segmentation approaches when the final task is to estimate a parameter, i.e., a biomarker, from the image. Although segmentation-based methods provide interpretable results for the HC estimation because the segmentation result is visible, they often require dedicated post-processing steps. On the other hand, regression approaches based on CNN are end-to-end, less costly, and prone to error and even though they do not offer explicit interpretability, this aspect can be explored using saliency maps for example [42]. In our paper, we explored both segmentation and segmentation-free approaches with state-of-the-art CNN architectures and backbones. By setting the same experimental conditions, we proposed a fair, quantitative comparison of these two approaches, in order to assess if the direct estimation approach is viable for this task. Even though the estimation error is much higher with the regression networks, the results are still promising and in line with inter-operator variability. Therefore, direct estimation, regression-based approaches have a high potential that should be deepened in the future. While we used general-purpose architectures for our regression methods, it would be interesting to investigate customized architecture for this task, and that includes attention mechanisms.

In future work, we will assess the generic regression CNNs on other medical datasets to estimate multiple biomarkers. Besides, we plan to investigate the segmentation-free approaches with other, recent CNN architectures that have a higher ability regarding feature representation, e.g., transformer architectures, as well as multi-task learning which combines a segmentation branch and regression branch.

## Figures and Tables

**Figure 1 jimaging-08-00023-f001:**
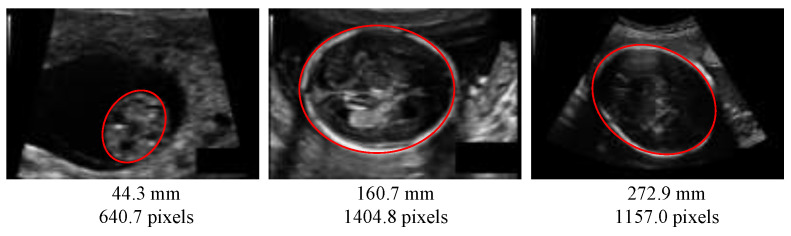
Ultrasound images of fetus head from the HC18 dataset [1] at different pregnancy stages. Red ellipses are head contours. Below the image, the corresponding head circumference (HC) is given. Images may have a different pixel size.

**Figure 2 jimaging-08-00023-f002:**
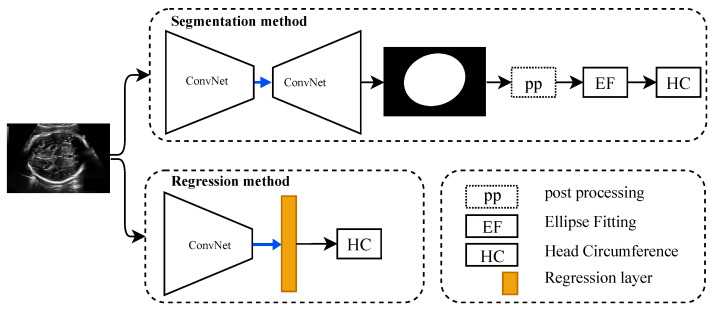
Overview of head circumference estimation process based on either segmentation-based method or regression-based method. HC: Head circumference, pp: Post-processing (the dotted box means is optional), and EF: Ellipse fitting.

**Figure 3 jimaging-08-00023-f003:**
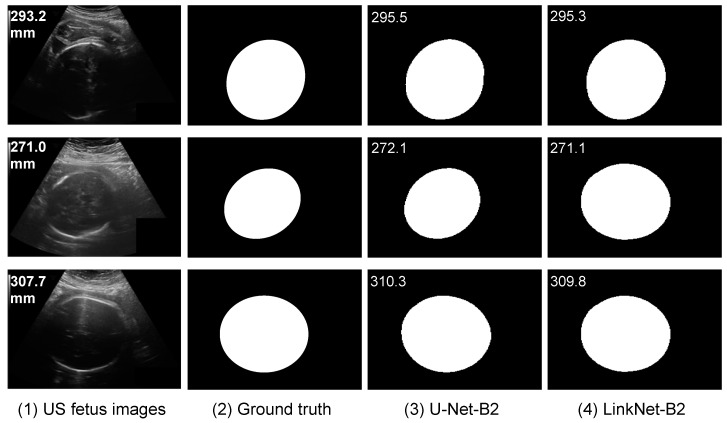
Segmentation results on three large fetus head and vague US images with U-Net-B2 and LinkNet-B2.

**Figure 4 jimaging-08-00023-f004:**
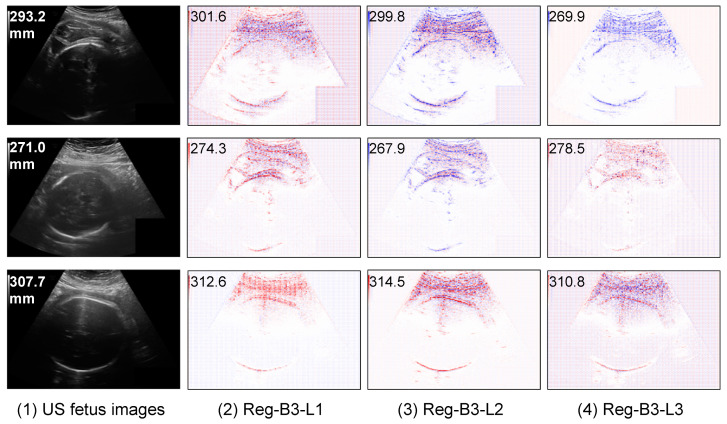
Saliency maps of regression CNN models on three cases of bad prediction results. The red points mean positive contribution and the blue points mean a negative contribution.

**Figure 5 jimaging-08-00023-f005:**
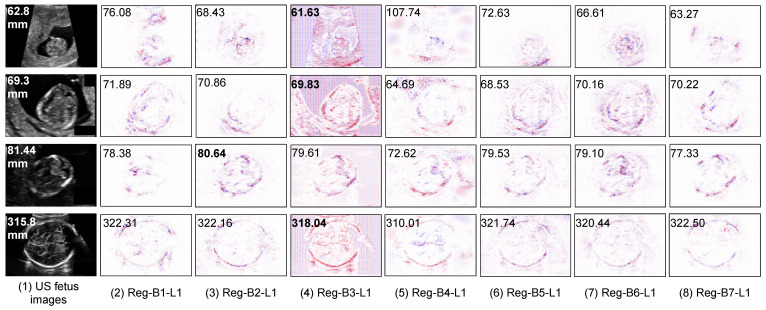
Saliency maps of different regression CNNs explained by LRP method. The numbers in input images and saliency maps are the ground truth and predicted HC values respectively. The best predicted results are in bold. The red points in saliency maps are positive values, the blue points are negative values.

**Figure 6 jimaging-08-00023-f006:**
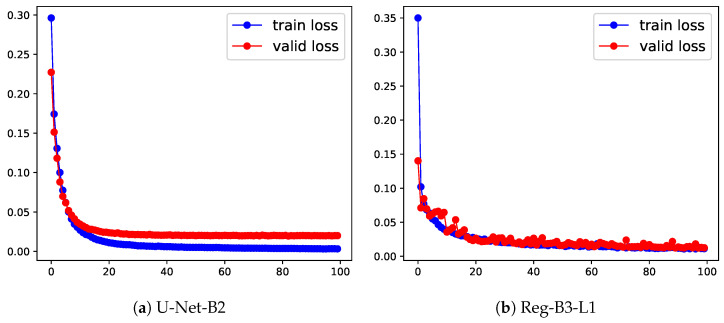
Learning curves of segmentation (U-Net-B2) vs. segmentation-free method (Reg-B3-L1) in the training and validation stage. The x-axis represents the training epochs; the y-axis is the loss.

**Figure 7 jimaging-08-00023-f007:**
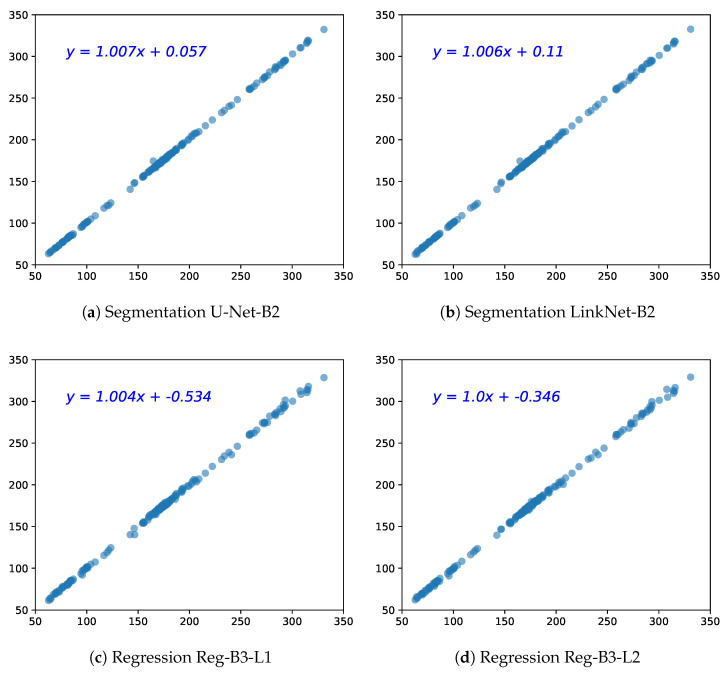
Scatter plots of the two best segmentation models U-Net-B2 and LinkNet-B2, and regression models (L1 = MAE loss, L2 = MSE loss). The x-axis represents the ground truth HC and the y-axis the predicted HC (in mm).

**Figure 8 jimaging-08-00023-f008:**
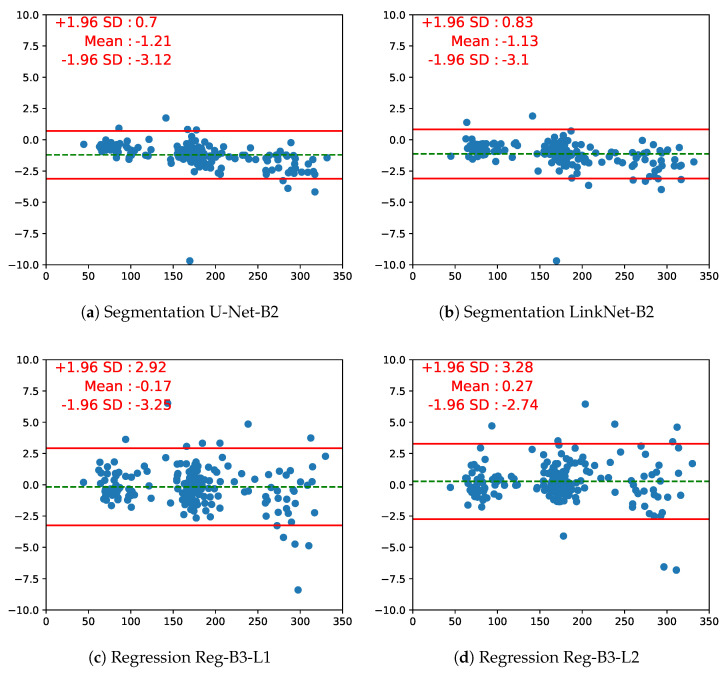
Bland–Altman plots of the segmentation (pretrained on ResNet50) and regression EfficientNet (efn) models (L1 = MAE loss, L2 = MSE loss). The x-axis represents the average value of ground truth and predicted HC; the y-axis represents the difference between ground truth and predicted HC (in mm). The horizontal red solid lines represent the upper and lower limits of 95% consistency. The middle dotted green line represents the mean of the difference.

**Table 1 jimaging-08-00023-t001:** Number of trainable parameters (#) of segmentation and regression CNN (convolutional neural network) models. M = million. Backbone names: VGG16 (B1), ResNet50 (B2), EfficientNetb2 (B3), DenseNet121 (B4), Xception (B5), MobileNet (B6), and InceptionV3 (B7). Reg = Regression.

Segmentation Models	# Parameters (M)	Regression Models	# Parameters (M)
Original U-Net	31.06	Reg-B1	15.15
U-Net-B1, B2, B3	23.75, 32.51, 14.23	Reg-B2	23.63
DoubleU-Net	29.29	Reg-B3	76.73
U-Net++ B1, B2, B3	24.15, 34.34, 16.03	Reg-B4	70.04
FPN-B1, B2, B3	17.59, 26.89, 10.77	Reg-B5	20.91
LinkNet-B1, B2, B3	20.32, 28.73, 10.15	Reg-B6	3.26
PSPNet-B1, B2, B3	21.55, 17.99, 9.41	Reg-B7	21.82

**Table 2 jimaging-08-00023-t002:** Segmentation accuracy of segmentation models and HC estimation accuracy with (w) and without (w/o) post-processing (pp). The results are given as mean and ±standard deviation. B1:VGG16, B2: ResNet50, B3: EfficientNetb2. DI = Dice index, HD = Hausdorff distance, ASSD = Average symmetric surface distance (mm), MAE = Mean absolute error (mm, pixel), PMAE = Percentage MAE. The best results are in bold.

Method	DI ↑ (%)	HD ↓ (mm)	ASSD ↓ (mm)	MAE ↓ (mm) w/o pp	MAE (mm) w pp	MAE (px) w/o pp	MAE (px) w pp	PMAE ↓ (%) w/o pp	PMAE (%) w pp
U-Net-original	98.5 ± 1.3	1.56 ± 2.67	0.35 ± 0.29	1.55 ± 4.41	1.23 ± 1.49	11.83 ± 38.75	9.11 ± 10.70	1.04 ± 4.13	0.75 ± 1.04
DoubleU-Net	98.7 ± 1.4	1.14 ± 0.86	0.29 ± 0.26	2.60 ± 1.89	2.59 ± 1.88	18.94 ± 11.53	18.76 ± 10.97	1.58 ± 1.19	1.56 ± 1.17
U-Net-B1	98.6 ± 1.3	1.16 ± 1.24	0.31 ± 0.26	1.31 ± 2.07	1.21 ± 1.29	9.99 ± 18.73	8.98 ± 8.48	0.85 ± 1.98	0.74 ± 0.75
**U-Net-B2**	98.8 ± 0.9	**1.09** ± 1.11	**0.27** ± 0.22	**1.16** ± 1.78	**1.08** ± 1.25	**8.69** ± 14.41	**7.87** ± 7.51	**0.74** ± 1.46	**0.65** ± 0.68
U-Net-B3	98.7 ± 1.1	1.11 ± 1.03	0.29 ± 0.24	1.34 ± 1.97	1.32 ± 1.67	10.23 ± 16.98	9.94 ± 13.58	0.86 ± 1.62	0.84 ± 1.32
U-Net++ B1	98.5 ± 2.4	1.29 ± 1.46	0.31 ± 0.25	2.03 ± 8.39	1.3 ± 2.12	16.95 ± 77.93	9.92 ± 18.52	1.51 ± 7.72	0.87 ± 2.32
U-Net++ B2	98.7 ± 1.0	1.24 ± 1.66	0.29 ± 0.23	1.74 ± 6.38	1.15 ± 1.59	12.65 ± 41.16	8.63 ± 12.24	1.16 ± 4.65	0.72 ± 1.13
U-Net++ B3	98.7 ± 1.2	1.17 ± 1.28	0.29 ± 0.25	2.32 ± 11.80	1.19 ± 1.44	19.08 ± 108.01	8.91 ± 11.01	1.57 ± 9.02	0.76 ± 1.21
FPN-B1	98.6 ± 1.1	1.28 ± 1.68	0.32 ± 0.27	1.44 ± 2.42	1.29 ± 1.61	11.17 ± 22.67	9.70 ± 12.84	0.99 ± 2.58	0.80 ± 1.16
FPN-B2	98.7 ± 0.9	1.18 ± 1.18	0.30 ± 0.23	1.38 ± 2.16	1.26 ± 1.33	10.35 ± 17.99	9.19 ± 8.58	1.90 ± 9.68	0.76 ± 0.87
FPN-B3	98.7 ± 1	1.19 ± 1.52	0.30 ± 0.25	1.46 ± 1.92	1.39 ± 1.5	11.09 ± 16.01	10.33 ± 10.52	0.94 ± 1.58	0.86 ± 1.06
LinkNet-B1	98.6 ± 1.2	1.31 ± 1.54	0.33 ± 0.25	1.46 ± 1.91	1.32 ± 1.44	11.32 ± 16.14	9.91 ± 10.23	0.98 ± 1.70	0.83 ± 1.02
**LinkNet-B2**	98.7 ± 1.1	1.12 ± 0.99	0.30 ± 0.23	**1.19** ± 1.56	**1.15** ± 1.32	**8.86** ± 11.83	**8.45** ± 8.39	**0.73** ± 1.08	**0.69** ± 0.77
LinkNet-B3	98.6 ± 1	1.15 ± 1.04	0.31 ± 0.26	1.37 ± 1.94	1.29 ± 1.51	10.55 ± 15.97	9.70 ± 9.84	0.89 ± 1.62	0.79 ± 0.84
PSPNet-B1	98.6 ± 1.4	2.01 ± 3.88	0.38 ± 0.44	3.07 ± 12.89	1.32 ± 1.38	22.38 ± 79.36	9.84 ± 9.03	2.21 ± 8.94	0.81 ± 0.81
PSPNet-B2	98.8 ± 0.9	1.42 ± 2.31	0.31 ± 0.28	1.66 ± 3.62	1.20 ± 1.34	11.98 ± 22.70	8.75 ± 7.98	1.07 ± 2.47	0.72 ± 0.68
PSPNet-B3	98.7 ± 1.1	1.12 ± 1.12	0.32 ± 0.25	1.38 ± 1.95	1.29 ± 1.36	10.59 ± 16.40	9.64 ± 9.14	0.93 ± 1.94	0.81 ± 0.86

**Table 3 jimaging-08-00023-t003:** Average performance of 21 regression CNN models over five-fold cross validation. The results are mean and ±standard deviation. MAE = Mean absolute error, PMAE = Percentage MAE. B1 = VGG16, B2 = ResNet50, B3 = EfficientNetb2, B4 = DenseNet121, B5 = Xception, B6 = MobileNet, B7 = InceptionV3, L1 = MAE loss, L2 = MSE loss, and L3 = Huber loss.

Model	MAE (mm)	MAE (px)	PMAE (%)
Reg-B1-L1	3.04 ± 2.97	22.41 ± 19.94	1.94 ± 2.19
Reg-B2-L1	3.24 ± 3.31	24.11 ± 22.65	2.14 ± 2.61
**Reg-B3-L1**	**1.83** ± 2.11	**13.57** ± 13.53	**1.17** ± 1.43
Reg-B4-L1	12.59 ± 12.49	93.63 ± 83.53	8.68 ± 11.25
Reg-B5-L1	2.96 ± 2.79	22.39 ± 19.34	1.89 ± 1.97
Reg-B6-L1	3.23 ± 3.29	24.29 ± 22.11	2.13 ± 2.50
Reg-B7-L1	3.34 ± 3.49	26.04 ± 27.89	2.28 ± 2.99
Reg-B1-L2	3.16 ± 3.28	23.83 ± 23.13	2.13 ± 2.69
Reg-B2-L2	3.73 ± 3.48	28.41 ± 26.99	2.55 ± 3.15
**Reg-B3-L2**	**2.35** ± 2.74	**17.32** ± 17.95	**1.53** ± 2.02
Reg-B4-L2	5.69 ± 5.92	43.54 ± 44.89	3.87 ± 4.97
Reg-B5-L2	3.12 ± 3.07	23.77 ± 22.19	1.99 ± 2.27
Reg-B6-L2	4.68 ± 4.17	35.39 ± 30.59	3.10 ± 3.36
Reg-B7-L2	4.33 ± 4.67	32.29 ± 32.60	2.87 ± 3.78
Reg-B1-L3	3.37 ± 3.72	25.75 ± 26.36	2.33 ± 3.05
Reg-B2-L3	3.12 ± 2.97	24.03 ± 23.69	2.11 ± 2.66
**Reg-B3-L3**	**2.78** ± 3.03	**20.62** ± 20.22	**1.79** ± 2.13
Reg-B4-L3	9.15 ± 9.07	70.49 ± 67.38	6.20 ± 7.39
Reg-B5-L3	3.40 ± 3.09	26.08 ± 21.34	2.19 ± 2.28
Reg-B6-L3	4.30 ± 4.44	32.48 ± 32.45	2.86 ± 3.67
Reg-B7-L3	6.29 ± 13.86	48.39 ± 111.02	4.33 ± 11.25

**Table 4 jimaging-08-00023-t004:** Comparison of HC estimation for the two best segmentation and regression (segmentation-free) models. B2: Resnet50. B3: EfficientNet, L1 = MAE loss, and L2 = MSE loss. The results are mean and ±standard deviation. MAE = Mean absolute error, PMAE = Percentage MAE. The best results are in bold. (*p* value < 0.05).

Metrics	MAE (mm)	MAE (px)	PMAE (%)
Methods	Segmentation-based methods		
**U-Net-B2**	**1.08** ± **1.25**	**7.87** ± **7.51**	**0.65** ± **0.68**
LinkNet-B2	1.15 ± 1.32	8.45 ± 8.39	0.69 ± 0.77
	Segmentation-free methods		
**Reg-B3-L1**	**1.83** ± **2.11**	**13.57** ± **13.53**	**1.17** ± **1.43**
Reg-B3-L2	2.35 ± 2.74	17.32 ± 17.95	1.53 ± 2.02

**Table 5 jimaging-08-00023-t005:** Training and predicting time and memory cost of segmentation vs. segmentation-free models on test set (200 images). B1 = VGG16, B2 = ResNet50, B3 = EfficientNetb2, B4 = DenseNet121, B5 = Xception, B6 = MobileNet, B7 = InceptionV3, L1 = MAE loss, Mem-M = theoretical memory of model, Mem-P = memory in prediction stage, and GB = gigabyte.

Methods	Train (s/Epoch)	Predict (s/Test Set)	Mem-M (GB)	Mem-P (GB)
	Segmentation-based methods			
U-Net-B2	29	68.26	3.06	1.84
DoubleU-Net	70	114.21	7.21	2.40
U-Net++-B2	68	172.45	7.26	2.34
FPN-B2	44	101.30	5.47	2.04
LinkNet-B2	30	80.36	3.82	1.90
PSPNet-B2	88	225.38	11.06	4.04
	Segmentation-free method			
Reg-B1-L1	17	30.86	0.96	1.36
Reg-B2-L1	20	48.28	2.31	1.73
Reg-B3-L1	38	36.95	2.29	2.68
Reg-B4-L1	21	65.55	3.01	1.69
Reg-B5-L1	35	51.78	2.15	1.67
Reg-B6-L1	14	18.71	1.03	1.14
Reg-B7-L1	17	22.55	1.09	1.60

**Table 6 jimaging-08-00023-t006:** Comparison of HC estimation with state-of-the-art on the HC18 dataset. B2 = ResNet50, B3 = EfficientNetb2, L1 = MAE loss, DI = Dice index, and N/A = Not applicable.

Metrics	MAE (mm)	DI (%)
Methods	Segmentation-based methods	
Budd et al. [16]	1.81 ± 1.65	98.20 ± 0.80
Sobhaninia et al. [18]	2.12 ± 1.87	96.84 ± 2.89
Fiorentino et al. [19]	1.90 ± 1.76	97.75 ± 1.32
Moccia et al. [20]	1.95 ± 1.92	97.90 ± 1.11
**U-Net-B2 (Proposed)**	**1.08** ± 1.25	**98.80** ± 0.9
	Segmentation-free methods	
**Reg-B3-L1 (Proposed)**	**1.83** ± 2.11	N/A

## Data Availability

We used the public HC18 dataset from https://hc18.grand-challenge.org/ (accessed on 11 December 2019).

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
