# Peer review of "Segmentation-Based vs. Regression-Based Biomarker Estimation: A Case Study of Fetus Head Circumference Assessment from Ultrasound Images"

_2313-433X, 2022, doi:10.3390/jimaging8020023_

Round 1

Reviewer 1 Report

Summary:

In this study, for the estimation of fetal head circumference, Authors proposed a fair, quantitative comparison of segmentation-based and segmentation-free (i.e. regression) approaches to estimate how far regression-based approaches stand from segmentation approaches.The workload of the article is very large and the experimental arrangement is reasonable.

General concept comments:

There are some problems in the conclusions and experimental design, I can feel that the author prefers the second method to the conclusion of the article,However, it is not convincing to show the potential of the second method only through the characteristic diagram.(Because you can also look at the split feature graph, maybe it's better). In addition, I hope to supplement the experiments of the two methods on standard test data(n=355), so that the paper will be more fair and reliable, Specific comments are as follows.

Reviewer 2 Report

The paper entitled „Segmentation-based vs. Regression-based Biomarker Estimation: A Case Study of Fetus Head Circumference Assessment from Ultrasound Images” by Zhang et al. deals with a segmentation method in order to automatically measure the head circumference of the fetus.

The authors follow their previous study from [42] (Interpretable and Annotation-Efficient Learning for Medical Image Computing; Springer, 2020) and propose in the current manuscript six networks used in medical images for segmentation of the contour of the fetus head.

The performance of the results are compared segmentation-free vs. the segmentation approaches, for method U-Net-B2 the MAE(mm and px) and PMAE values exceed the obtained value for LinkNet-B2, the same thing happens with Reg-B3-L1 and Reg-B3-L2.

In addition, the paper is well-written and carefully organized.

The results are compared with new references [16, 18-20].

The conclusions support the results.

There are also some editorial issues that should be revised.

Eq. (4), what is parameter δ?

Line 189: add Table 2

In the legend of table 1 please add the association of the Backbone names between brackets: VGG16 (B1), and so on.

A few conceptualization issues.

In figure 6 it is a scatter plots of the 2 best segmentation models U-Net-B2 and LinkNet-B2 in mm, what is the trend when work in px.

In the acquisition stage the US images are affected by noise and artifacts, please specify if these unwanted elements influence the proposed segmentation method.  

Round 2

Reviewer 1 Report

Thank you very much for your reply and looking forward to see your further exploration of the related work.